# Detection of Crimean-Congo Hemorrhagic Fever Virus Antibodies in Cattle in Plateau State, Nigeria

**DOI:** 10.3390/v14122618

**Published:** 2022-11-24

**Authors:** Asabe A. Dzikwi-Emennaa, Clement Meseko, Paulinus Emennaa, Adedeji J. Adeyinka, Andrew M. Adamu, Oyelola A. Adegboye

**Affiliations:** 1Department of Veterinary Public Health and Preventive Medicine, University of Jos, Jos 930003, Nigeria; 2National Veterinary Research Institute, Vom, Jos 930101, Nigeria; 3Department of Veterinary Public Health and Preventive Medicine, University of Abuja, Abuja 900105, Nigeria; 4Australian Institute of Tropical Health and Medicine, James Cook University, Townsville, QLD 4811, Australia; 5College of Public Health, Medical and Veterinary Sciences, James Cook University, Townsville, QLD 4811, Australia; 6World Health Organization Collaborating Center for Vector-Borne and Neglected Tropical Diseases, College of Public Health, Medical and Veterinary Sciences, James Cook University, Townsville, QLD 4811, Australia

**Keywords:** cattle, CCHF, ELISA, Nigeria, antibodies

## Abstract

Crimean-Congo hemorrhagic fever (CCHF) is a vector-borne viral hemorrhagic disease with global clinical significance. Certain species of ticks are vectors of CCHF, which can be transmitted from animals to humans and humans to humans by direct exposure to blood or other body fluids. The zoonotic transmission at the human–animal interface from viremic animal hosts to humans is a public health concern with a paucity of data in Nigeria. Samples from 184 pastoral cattle from three local government areas (LGAs) of Plateau state, Nigeria, were screened for CCHF virus using a commercial enzyme-linked immunosorbent assay (ID Screen^®^ CCHF Double Antigen for Multi-Species). Overall seropositivity of 30.4% (*n* = 56) (95% CI: 23.88%, 37.63%) was recorded from the study areas in Plateau State, while 48/126 (38.1%, 95% CI: 29.59%, 47.17%) sampled cows tested positive for CCHFV antibodies. Seropositivity was significantly higher (*p* < 0.001) among older cattle greater than two years, 54.69% (95% CI: 2.88%, 11.24%) compared to cattle younger than two years, 17.5% (95% CI: 11.17%, 25.50%). The location of farms played a significant role in the seropositivity of CCHF with the least risk observed in Wase LGA. CCHF is an important zoonotic disease in different parts of the globe with a high risk of transmission to pastoralists, livestock keepers/slaughterhouse workers, and veterinarians who handle animals. There is a need for a collaborative one-health approach with various stakeholders to unravel the dynamics of CCHFV epidemiology in Nigeria.

## 1. Introduction

Crimean-Congo hemorrhagic fever (CCHF) is an arboviral zoonosis of humans and livestock caused by the CCHF virus belonging to the genus *Orthonairovirus* and the family *Nairoviridae* [1,2]. CCHF is an important yet neglected disease with severe public health and socio-economic consequences. The World Organization for Animal Health and the World Health Organization prioritized Crimean-Congo hemorrhagic fever virus (CCHFV) as one of the leading emerging and re-emerging pathogens of interest. Clinical disease is restricted to humans, and its case fatality rates vary; in 3–100% of cases [3,4,5,6]. CCHF occurs over a wide geographic area, including Asia, Africa, and Europe [7,8]. The natural vectors have been identified as ticks such as *Hyalomma* spp., *Ambyomma* spp., *Rhipicephalus* spp. [9], which are predominant in herds of cattle in Nigeria [10], and the distribution of human cases closely mirror vector distribution [8].

In an earlier serological study, Causey et al. [10] revealed the presence of several arboviruses in Nigeria, including CCHFV. Limited serologic evidence of CCHFV infection has been reported in ruminants in Nigeria [11]. It is well documented that viremia and CCHFV-specific antibodies develop in infected livestock, including sheep, goats, cattle, and camels [12]. Infected livestock may infect humans thus playing an important role in the epidemiology of the disease [13,14]. A wide distribution of livestock in Nigeria and increased transhumance due to climate change have had a major impact on the global emergence and re-emergence of arthropod-borne viral diseases [15].

We carried out this study in three local government areas (LGAs), Plateau State, North-Central, Nigeria. The state shares border with Bauchi State, which has one of the largest game reserves in the country (Yankari Game Reserve) and many wildlife parks where Rift Valley fever virus antibodies have been reported in wildlife and domestic animals [16]. Therefore, in this study, we aimed to carry out seropositivity of the CCHF virus in Plateau. The study provides crucial data on the seroprevalence of CCHFV, one of the priority pathogens in Plateau State that has a conducive environment for cattle rearing.

## 2. Materials and Methods

### 2.1. Study Area

Plateau State is located in North-Central Nigeria, between latitude 9.1667–9.2182° N and longitude 9.5179–9.7500° E (Figure 1). Plateau State has one of the highest altitudes in Nigeria, with a near-temperate climate and with a temperature range of 13 °C to 22 °C. The Plateau records 146 cm of rainfall compared to the southern senatorial part of the state with 136 cm annual mean rainfall.

### 2.2. Study Design and Sample Size

A total of 184 cattle were selected randomly using a multistage probability sampling method from cattle herds in two LGAs in the southern senatorial district and one LGA in the central senatorial district of Plateau State (Figure 1). Using the formula described by Thrusfield [17], the sample size of 184 was estimated based on a prevalence of 10% [8] to 24% [11] and a margin of error of 10%.

### 2.3. Data Collection

Blood samples were collected from the jugular vein using an 18 G needle and a 10 mL syringe to collect 5 mL of blood from the transhumance cattle herds in the three LGAs of Plateau State. The blood was labelled appropriately in a 5 mL tube and transported on ice packs to the laboratory. After centrifugation at 5000× *g* for 10 min, two milliliters of sera were collected. The samples were kept at −20 °C until used for detection of CCHFV-specific Ig G antibodies at the National Veterinary Research Institute, Vom, Plateau State, Nigeria.

### 2.4. Enzyme-Linked Immunosorbent Assay (ELISA)

The sera were analysed for antibodies to CCHF using a commercial ELISA kit (ID Screen^®^ CCHF Double Antigen for Multi-Species I.D Vet, Grabels, France) to detect specific antibodies against the nucleoprotein (NP) of the CCHF virus in ruminants and other susceptible species according to the manufacturer’s recommendation. Freeze-dried recombinant CCHFV soluble protein antigen was linked to horseradish peroxidase (HRP) as conjugate [18]. Anti-NP antibodies present in the serum formed an antibody–antigen complex in a solid phase reaction. Briefly, 50 μL dilution buffer was added to all wells of a flat-bottomed microtiter plate pre-coated with 5 μg/μL recombinant and purified CCHF nucleoprotein. Thereafter, positive and negative controls were dispensed into each designated well while 30 μL test sera were added to the rest of the wells. The plates were incubated for 45 min at 21 °C. After that, the plate was washed five times with a 300 μL Washing Buffer. Thereafter, 50 μL of ×10 horseradish peroxidase conjugate diluted at 1:10 was added to all wells. Following a 30 min incubation at 21 °C, the plate was again washed five times with 300 μL washing buffer. One hundred μL of substrate solution (Tetramethylbenzidine) was subsequently added to each well, and another incubation was carried out for 15 min in the dark. Finally, 100 μL Stop Solution was added to each well to stop the reaction.

Measurement of the optical density (OD) was done using a spectrophotometer (Thermoscientific™ Multiskan™, Waltham, MA, USA) at 450 nm. For each sample, the S/P% was calculated as the ratio of the optical density to the calculated mean of the positive control multiplied by 100. Serum samples with S/P% ≤ 30% were classified as negative, while samples with S/P% > 30% were considered positive according to the manufacturer’s recommendation.

### 2.5. Data Analysis

Data were analyzed in R version 3.6.1 (https://www.R-project.org). Sample characteristics (ELISA result, age, sex, and location) were presented as frequencies and percentages. The Clopper-Pearson intervals were presented for the proportions, while the chi-square test was used for testing the equality of proportions. Multivariable logistic regression was used to investigate the risk factors associated with the seropositivity of CCHFV. The strength of the association between categorical variables were presented as adjusted odds ratios (aOR) and 95% confidence interval (CI). A 5% level of significance was set for the study.

## 3. Results

Of the 184 cattle sampled, the majority, 126 (68.49%), were female (cows), and the rest, 58 (31.52%), were male cattle (bulls) (Table 1). The number of samples from each of the three LGA were fairly similar: 55 (29.89%) in Kanam, 64 (34.78%) in Shendam, and 65 (35.33%) in Wase. A higher proportion of the cattle were 0–2 years old (120, 65.22%), while 64 (34.78%) were 3+ years. The prevalence of CCHFV was *n* = 56/184 (30.43%, 95% CI: 23.88%, 37.63%) in the study area (Table 1). More cows tested positive for CCHFV antibodies than bulls (38.1%, 95% CI: 29.59%, 47.17% vs. 13.79%, 95% CI: 6.15%, 25.38%). The highest prevalence of CCHFV antibodies was recorded in Shendam LGA at 48.4% (85% CI: 35.75%, 61.27%) followed by Kanam LGA at 32.7% (95% CI: 35.75%, 61.27%) (*n* = 18/55), and the least prevalence was in Wase LGA at 10.8% (95% CI: 4.44%, 20.94%). Higher seropositivity of 54.69% (*n* = 35/64) was observed among older cattle greater than two years, recorded, and was found to be over five times (aOR = 5.69, 95% CI: 2.88, 11.24) more likely to be infected with CCHFV after adjusting for other variables in the model. Cows were about three and a half times more likely to be infected with CCHFV than bulls (aOR = 3.49; 95% CI: 1.53, 7.97). The risk of seropositivity was the least in Wase LGA (aOR = 0.25; 95% CI: 0.09, 0.65) compared to Kanam LGA.

## 4. Discussion

In this study, we demonstrated that cattle in Plateau State, Nigeria, were previously exposed to CCHFV, with an overall prevalence of 30.4%. The prevalence varied geographically across the study region. For example, cattle from Shendam LGA of Plateau State had the highest seropositivity compared to other LGAs. We attributed this to the presence of the Yelwa Shendam livestock market in the area where cattle conglomerate for sale from adjoining LGAs and the neighbouring Taraba State (which shares a border with Cameroon). In addition, most cattle are transported on hooves to markets through the bush where they are in proximity to different reservoirs of vector-borne diseases and wildlife; therefore, this may have played a role in the prevalence observed.

Our study revealed that the Wase local government area had the least significant risk based on location. The Nigerian government earmarked a grazing reserve in this location due to the fertile land, which has a lot of pasture for livestock grazing. This local government also has an area that is conserved for wildlife development and serves as a bird sanctuary (rosy white pelicans). The role of these birds in the dynamics of CCHFV is unknown. However, studies have demonstrated birds’ roles in spreading CCHFV in Morocco and Turkey [4,19]. The detection of antibodies to CCHFV in cattle in the study areas may have identified the nucleus of CCHFV and highlighted possible places for human infection, which could be investigated in detail [20]. The presence of CCHFV antibodies in cattle corroborates a recent study in Nigeria [10] where cattle play a role in the epidemiological dynamics and persistence of CCHFV in an environment.

Similarly, Obanda and colleagues [21] observed that the overlap between cattle and buffalo makes cattle a super spreader host (bridge species) of CCHFV and increases transmission risks to humans. The primary natural vector of CCHFV-*Hyalomma* spp. is believed to have an affinity for large ruminants [22]. Since the WHO and FAO prioritize CCHFV, and outbreaks are likely to be under-reported or misdiagnosed in resource-poor health facilities, there is an urgent need to scale up surveillance in both humans and animals in Plateau State and Nigeria.

We found risk factors such as age and sex to be significantly associated with CCHFV prevalence and a higher prevalence among older cattle than young cattle. The older cattle are believed to be exposed to many environmental and ecological risk factors that maintain *Hyalomma* spp. and other amplifying virus hosts, such as hedgehogs and birds, which could transmit the virus mechanically [22]. Young animals are generally given more veterinary care and may not travel long distances in search of feed; they have less exposure to predisposing factors of CCHFV, in contrast to the older animals that have had more exposure to ticks. On gender, cows had higher seropositivity than bulls. Cows and females are kept longer in a herd for breeding purposes, whereas males are fattened and sold. This risk factor is important as it serves as a good avenue for viral transmission, especially among veterinarians and livestock attendants during assisted delivery. Although there are scanty reports of sexual transmission of CCHFV [17], it calls for concern in animal production where the virus can spread to the fetus (trans-placental).

Our study is not without limitation as the authors did not employ molecular techniques for the detection of the viral genome of the sampled animals. This limitation may have led to an underestimate of CCHFV presence. Despite this limitation, we believe that the serosurvey gave insight into the prevalence of CCHFV in Nigeria.

## 5. Conclusions

In summary, in this study, we established the presence of CCHFV antibodies in cattle in Plateau State for the first time as one of the few studies in northern Nigeria. There is a need to increase surveillance in both human and livestock populations across Plateau state and Nigeria using the One Health approach with various stakeholders. This emphasizes intersectoral collaboration involving medical, veterinary, and environmental services to map out the areas of vector distribution. CCHFV is an arbovirus, hence, there is a need for advocacy on the use of acaricide for effective vector control and awareness among high-risk exposure groups such as cattle herders, veterinarians, and butchers.

## Figures and Tables

**Figure 1 viruses-14-02618-f001:**
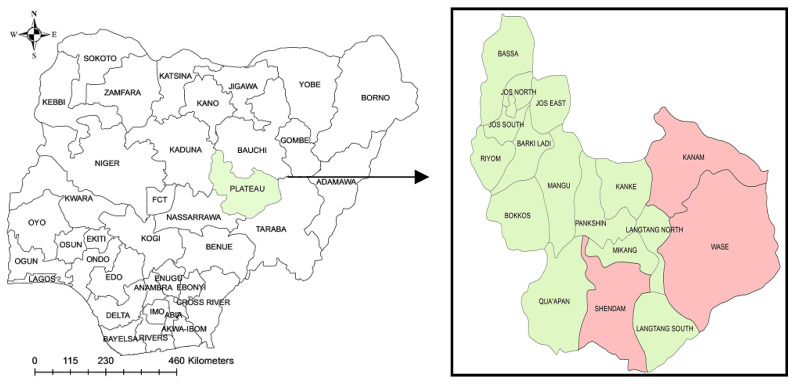
Map of Nigeria showing Plateau State in green. The study regions (three local governments) are in pink colour.

**Table 1 viruses-14-02618-t001:** Distribution of CCHFV antibodies in cattle in Plateau State, Nigeria.

Variables			Positive			Negative			*p*-Value ^b^	aOR (95% CI)
	n	%	n	%	95% CI ^a^	n	%	95% CI ^a^		
Seropositive	184	100	56	30.43	23.88, 37.63	128	69.57	62.37, 76.12		
Age, years										
0–2	120	65.22	21	17.50	11.17, 25.50	99	82.50	74.50, 88.83	<0.0001	Ref
≥3	64	34.78	35	54.69	41.75, 67.18	29	45.31	32.82, 58.25		5.69 (2.88, 11.24)
Sex										
Male	58	31.52	8	13.79	6.15, 25.38	50	86.21	74.62, 93.85	0.002	Ref
Female	126	68.49	48	38.10	29.59, 47.17	78	61.90	52.83, 70.41		3.49 (1.53, 7.97)
Location										
Kanam	55	29.89	18	32.73	20.68, 46.71	37	67.27	53.29, 79.32	<0.0001	Ref
Shendam	64	34.78	31	48.44	35.75, 61.27	33	51.56	38.73, 64.25		1.93 (0.92, 4.04)
Wase	65	35.33	7	10.77	4.44, 20.94	58	89.23	79.06, 95.56		0.25 (0.09, 0.65)

^a^ 95% CI based on Clopper-Pearson intervals. ^b^ Test of equality of proportions of positive CCHFV antibodies with a group. Ref: Reference category.

## Data Availability

The data presented in this article are available within the manuscript.

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
