# Peer review of "Detection of Crimean-Congo Hemorrhagic Fever Virus Antibodies in Cattle in Plateau State, Nigeria"

_viruses, 2022, doi:10.3390/v14122618_

Round 1

Reviewer 1 Report

The authors of manuscript designed their study to investigate the possibility of CCHF infection in Nigeria cattle. The authors collected cattle bloods from three different regions in Nigeria according to the age and gender. Their study results shows, when the authors tested the cattle blood serum using ELISA kit for CCHF specific IgG, total of 38.1% of cattle were positive for CCHFV antibody. The authors also further analyzed their data based on gender and age which the results show younger cattle were less lucky get infected compare to the adults and females (cow) were more lucky to get infected compared to the males (Bulls).

Overall the study designed nicely and the method is proper. The authors written the results clearly in sufficient details. The discussion, explain and discuss the results nicely and support the study finding. I have no comments for authors to improve their manuscript.

Author Response

Thank you

Reviewer 2 Report

In this manuscript, the authors have presented serological data to show presence of CCHFV antibodies in cattle from three local government areas (LGAs), in the Plateau State of Nigeria. Authors go on to argue that the presence of these antibodies alludes to the presence of the virus/virus circulating in the region. This can lead to a potential spillover into humans and poses an epidemic threat. Authors propose a One Health approach to better deal with this threat. Authors claim this to be the first study establishing the presence of CCHFV antibodies in cattle in Plateau State for the first time. While the study has merit and there is a need for this information to spread, here are some of the comments/concerns:

Comments:

11. Authors should discuss their motivation and perhaps the lack thereof, to focus on this state and restriction to these three local government areas (LGAs)?

22. Authors have sampled 184 cattle based on an assumption that the prevalence is 10% to 24% with a margin of error 10%. Where did this number come from?

SS3. Since approximately 30% of the cattle had antibodies, did authors try to detect viremia? There is a possibility that some of the negative animals may have been positive for the virus but had not developed antibodies yet? This will significantly change the urgency.

Author Response

1.Authors should discuss their motivation and perhaps the lack thereof, to focus on this state and restriction to these three local government areas (LGAs)?

Response: We thank the reviewer for their comments and suggestions and the opportunity to revise our manuscript. 

The rationale for the selection of the three LGAs presented in lines 55-50 as:

"We carried out this study in three local government areas (LGAs), Plateau State, North-Central, Nigeria. The state shares border with Bauchi State, which has one of the largest game reserves in the country (Yankari Game Reserve) and many wildlife parks where Rift Valley fever virus antibodies have been reported in wildlife and domestic animals [16]. Therefore, in this study, we aimed to carry out seropositivity of the CCHF virus in Plateau."

2.Authors have sampled 184 cattle based on an assumption that the prevalence is 10% to 24% with a margin of error 10%. Where did this number come from?

Response: These numbers are from previous studies cited in the text.

Using the formula described by Thrusfield [17], the sample size of 184 was estimated based on a prevalence of 10% [8] to 24% [11], and a margin of error of 10%.

Ref 8: Bukbuk DN, Dowall SD, Lewandowski K, Bosworth A, Baba SS, Varghese A, Watson RJ, Bell A, Atkinson B, Hewson R. Serological and Virological Evidence of Crimean-Congo Haemorrhagic Fever Virus Circulation in the Human Population of Borno State, Northeastern Nigeria. PLoS Negl Trop Dis. 2016, 7;10:e0005126. https://doi: 10.1371/journal.pntd.0005126.

Ref 11: Oluwayelu D, Afrough B, Adebiyi A, Varghese A, Eun-Sil P, Fukushi S, Yoshikawa T, Saijo M, Neumann E, Morikawa S, Hewson R, Tomori O. Prevalence of Antibodies to Crimean-Congo Hemorrhagic Fever Virus in Ruminants, Nigeria, 2015. Emerg Infect Dis. 2020, 26:744-747.https:// doi: 10.3201/eid2604.190354

3. Since approximately 30% of the cattle had antibodies, did authors try to detect viremia? There is a possibility that some of the negative animals may have been positive for the virus but had not developed antibodies yet? This will significantly change the urgency.

Response:

We did not detect viremia in this study. The authors received no funding for this project, and subsequent studies will include molecular techniques. 

Reviewer 3 Report

Introduction:

- page 1 line 41: define CCHFV (CCHF previously defined)

- page 1 line 41: replace "fatal" with "fatality"

- page 2 line 59: replace "study is" with "study provides"

Materials and methods:

- page 2 line 63: degree for latitude has an underscore

- page 2 figure 1: provide more information in the heading - ie which area, in which country

- page 2 line 70-71: remove this sentence - this is covered later on in the manuscript.

- page 2 line 78: replace "A total of 184 blood samples" with "Blood samples"

- page 3 line 84 - in which country is the National Veternary Research Institute, Vom located?

- page 3 line 99: replace word "microliters" with symbol

- page 3 line 104: define "S/P"

- page 3 line 105-107: positive and negative should be mutually exclusive - both cannot include 30%

- page 3 line 109: provide info for "R" - company or website

Results:

- page 3 line 117: recommend to start the results with a general descritpion of the study population.

- page 3 line 121-121: remove sentence "Instrinsic risk factors... logistic regression" - this is methods

- page 3 line 126-127: rephrase sentence "Based on sex...of bulls" - not reading clearly

- page 4 table 1: combine the aOR results with table 1 and remove figure 1 - does not add additional value 

Discussion:

- page 4 line 144: remove "of 48.4%"

- page 5 line 175: replace "high" with "higher"

Conclusion: 

- page 5 line 3: define what state are the authors referring to

Author Response

Introduction:

Que: page 1 line 41: define CCHFV (CCHF previously defined)

Response: This has been corrected.

Que: Page 1 line 41: replace "fatal" with "fatality"

Response: It has been replaced

Que: page 2 line 59: replace "study is" with "study provides"

Response: The phrase ‘study provides’ has been  inserted as recommended by the reviewer, line 60

Materials and methods:

Que: page 2 line 63: degree for latitude has an underscore

Response: Thank you, this has been corrected

Que:  page 2 figure 1: provide more information in the heading – ie  which area, in which country

Response: We have revised the caption as “Map of Nigeria showing Plateau State in green. The study region (three local governments) sampled are in pink colour.

Que: page 2 line 70-71: remove this sentence - this is covered later on in the manuscript.

Response: The sentence has been removed

Que: page 2 line 78: replace "A total of 184 blood samples" with "Blood samples"

Response: Blood samples have been inserted at the beginning of the sentence.

Que:  Page 3 line 84 - in which country is the National Veterinary Research Institute, Vom located?

Response: The country Nigeria has been included as “National Veterinary Research Institute, Vom, Plateau State, Nigeria.

Que:  page 3 line 99: replace the word "microliters" with the symbol

Response: The symbol has been used instead

Que: page 3 line 104: define "S/P"

We have defined S/P as follows: “For each sample, the S/P% was calculated as the ratio of the optical density to the calculated mean of the positive control multiplied by 100.”

Que: page 3 line 105-107: positive and negative should be mutually exclusive - both cannot include 30%.

Response: Lines 104-106 have been addressed and highlighted in red.

Que:  page 3 line 109: provide info for "R" - company or website

Response:  We have added the website. “https://www.R-project.org”

Results:

Que: page 3 line 117: recommend to start the results with a general descritpion of the study population.

Response: The results have been revised.

Que:  page 3 line 121-121: remove sentence "Instrinsic risk factors... logistic regression" - this is methods

Response: this methodology has been removed as suggested by the reviewer.

Que:  page 3 line 126-127: rephrase sentence "Based on sex...of bulls" - not reading clearly

Response: The sentences from lines 123-125 have been clarified \.

Que: page 4 table 1: combine the aOR results with table 1 and remove figure 1 - does not add additional value 

Response:  We have removed Figure 1 and revised Table 1.

Discussion:

Que: page 4 line 144: remove "of 48.4%"

Response: The authors have accepted the suggestion and have removed  48.8%

Que: page 5 line 175: replace "high" with "higher"

Response: high has been replaced with higher, line 174

Conclusion: 

Que: page 5 line 3: define what state are the authors referring to

Response:  We have reworded it as “There is a need to increase surveillance in both human and livestock populations across Plateau state and in Nigeria using the One Health approach with various stakeholders.”